# A Cluster-Based Energy Optimization Algorithm in Wireless Sensor Networks with Mobile Sink

**DOI:** 10.3390/s21072523

**Published:** 2021-04-04

**Authors:** Qian Wei, Ke Bai, Lin Zhou, Zhentao Hu, Yong Jin, Junwei Li

**Affiliations:** 1School of Artificial Intelligence, Henan University, Kaifeng 475004, China; weiqian@vip.henu.edu.cn (Q.W.); zhoulin@henu.edu.cn (L.Z.); hzt@henu.edu.cn (Z.H.); jy@henu.edu.cn (Y.J.); lijunwei@henu.edu.cn (J.L.); 2School of Computer and Information Engineering, Henan University, Kaifeng 475004, China

**Keywords:** mobile sink, energy optimization, cluster head selection, adaptive adjustment function

## Abstract

Aiming at high network energy consumption and data delay induced by mobile sink in wireless sensor networks (WSNs), this paper proposes a cluster-based energy optimization algorithm called Cluster-Based Energy Optimization with Mobile Sink (CEOMS). CEOMS algorithm constructs the energy density function of network nodes firstly and then assigns sensor nodes with higher remaining energy as cluster heads according to energy density function. Meanwhile, the directivity motion performance function of mobile sink is constructed to enhance the probability of remote sensor nodes being assigned as cluster heads. Secondly, based on Low Energy Adaptive Clustering Hierarchy Protocol (LEACH) architecture, the energy density function and the motion performance function are introduced into the cluster head selection process to avoid random assignment of cluster head. Finally, an adaptive adjustment function is designed to improve the adaptability of cluster head selection by percentage of network nodes death and the density of all surviving nodes of the entire network. The simulation results show that the proposed CEOMS algorithm improves the cluster head selection self-adaptability, extends the network life, reduces the data delay, and balances the network load.

## 1. Introduction

Wireless Sensor Networks (WSNs) are composed of thousands of sensor nodes, which are characterized by small size, low cost and low power consumption. Therefore, WSNs are widely used in military reconnaissance, climate and environment monitoring, natural disaster warning and treatment, intelligent medical technology and other fields [1,2,3]. WSNs can be divided into static wireless sensor networks (static WSNs) and mobile wireless sensor networks (mobile WSNs) according to the types of sensor nodes [4,5]. In the static WSNs, the location of sensor nodes is fixed once it’s deployed. Since nodes with limited energy nearby the static sink node may be assigned as cluster head frequently, leading to high energy consumption of those nodes, thus may cause premature death of those sensor nodes. In mobile WSNs, sensor nodes can move according to specific mission, thus mobile sink may be introduced to alleviate energy consumption of nodes nearby sink node [6,7,8,9,10].

Supposing links between mobile sink and other nodes in a time slot are unchangeable, mobile WSNs can be simplified to static WSNs in this time slot, thus efficient and energy-saving routing algorithms based on static WSNs may be extended to mobile WSNs [11,12,13]. The Low Energy Adaptive Clustering Hierarchy Protocol (LEACH) proposed by Heinzelman et al. [14,15] is the most classic hierarchical routing algorithm. This algorithm divided network nodes into different clusters firstly, and then used periodically replacing cluster heads to balance network Energy consumption, leading to network life cycle prolongation. In addition, A Hybrid, Energy-Efficient, Distributed Clustering Approach (HEED) [16], A Stable Election Protocol for Clustered (SEP) [17], and LEACH-Centralized (LEACH-C) [18] have also been proposed to reduce energy consumption, balance network resources, and extend the network life cycle.

### 1.1. Related Work

Since mobile sink may change topology of data transmission in mobile WSNs, using traditional routing algorithms of static WSNs in mobile WSNs may deteriorate energy balance of mobile WSNs, result in a short network life cycle. Some scholars proposed routing protocols which use mobile sink to balance energy consumption of network. According to distance with mobile sink or residual energy, a node may be assigned as a rendezvous point to collect data of nearby nodes and then to communicate with mobile sink [19,20]. Using mobile data collector and hierarchical clustering technique, Zhang et al. proposed a data collection routing algorithm which assigns node with maximum density as cluster head to avoid long-distance transmission energy consumption, leading to network resources balance [21]. Zhu et al., proposed a data collection algorithm based on tree clusters technique, which selected specific rendezvous points and sub-rendezvous points according to the remaining energy of the node and the number of multi-hops respectively, and collected the data of each rendezvous point by mobile sink [22]. Although using mobile sink leading to network life prolongation, above algorithms may introduce mutual interference of data transmission, resulting in data deviation and data delay. Moreover, randomly selecting cluster heads may cause some nodes be assigned as cluster heads frequently, leading these nodes to dead premature, thus degrading the performance of the whole network.

To tackle problems caused by mobile sink such as cluster head uneven distribution, data redundancy and delay, researchers proposed some clustering routing algorithms. Jing et al. proposed an improved LEACH protocol (ILEACH) [23] to select nodes as cluster heads according to remaining energy of node, thus enhance selected probability of nodes with high remaining energy. Sharma et al. also proposed an improved LEACH algorithm called Distance Based Cluster Head (DBCH) [24], which selects node as cluster head following some criterion such as distance between node and the sink node, the maximum and minimum distance between nodes and sink node, the remaining energy and so on. Integrating underlying factors related to energy balance of WSNs such as the remaining energy of the node, the distance between the node and the sink node, and the average distance of all nodes with the sink node together, Darabkh et al. proposed an improved algorithm called LEACH-Distance Based Thresholds (LEACH-DT) [25].

Obviously, reasonable use of information of location and velocity related to mobile sink to select cluster-head may contribute to balance energy consumption of WSNs [26]. Using mobile sink technique, Wang et al., proposed a stable election protocol based on improved SEP [11,17]. This algorithm classified sensor nodes firstly and then constructed threshold function of cluster head selection based on the remaining energy and initial energy of nodes. Kushal et al., clustered sensor nodes according to the location of nodes firstly and then assigned cluster head of each cluster according to the remaining energy of nodes and the distance between nodes [27]. Moreover, using location information of mobile sink broadcasted by itself, Kushal’s method constructs the shortest multi-hop between cluster head and mobile sink to reduce energy consumption of WSNs. We can find that above algorithms use remaining energy information and location information of nodes to balance distribution of cluster heads, thus extending life cycle of WSNs and reducing data transmission delay. However, supposing the velocity of the mobile sink is unchangeable, above algorithms may unsuitable in the scene of mobile sink variable-speed moving.

### 1.2. Contributions

Summarizing, the current energy-saving algorithms focus less on energy consumption and transmission delay caused by mobile sink moving in WSNs. This paper overall considers multi-factors related to energy balance of WSNs including remaining energy rate of WSNs, density of nodes, location changing of mobile sink and mortality rate of nodes to propose a new network energy optimization algorithm called Cluster-Based Energy Optimization with Mobile Sink (CEOMS). The contributions of this paper are as follows:(1)The energy density function containing variables such as residual energy rate and density nearby sensor nodes is constructed to assign nodes with high remaining energy as cluster heads. Compared with the traditional clustering algorithm, proposed algorithm fully considers the remaining energy of network nodes and the neighborhood density of nodes, thus extending the network life.(2)The motion performance function containing variables such as velocity of mobile sink, distance between mobile sink and node is constructed to enhance probability of the remote node be assigned as cluster head. Since fully considering underlying factors related to network energy balance such as moving distance and direction of mobile sink, the proposed algorithm has better ability to balance network energy than that of the traditional clustering algorithm.(3)The cluster head selection contains two independent functions, including energy density function and motion performance function. Moreover, the adaptive adjustment function is introduced to adjust the weight parameters of the energy density function and the motion performance function. The three functions constitute the adaptive cluster head selection threshold, which can avoid nodes’ premature decay, leading to network life extending.

### 1.3. Paper Organization

The content of this paper is organized as follows. Section 2 gives the network model of the algorithm and motion model of mobile sink; Section 3 elaborates principle and implementation process of the CEOMS algorithm; Section 4 verifies the feasibility and effectiveness of the CEOMS algorithm through simulation. Section 5 summarizes the paper.

## 2. System Model

Based on classical clustering routing LEACH protocol, the system model in WSNs with mobile sink is constructed in this paper. Besides, the motion model of mobile sink is also presented.

### 2.1. Network Model

WSNs composed of sensor nodes, mobile sink, satellites, Internet, control center and users is shown in Figure 1. This figure is a typical example of WSNs applied to smart urban planning [28,29]. The sensor nodes are randomly distributed in the monitoring area, and the mobile sink moves according to a specific motion model and collects data from all sensor nodes. However, when all sensor nodes send data directly to the mobile sink, some nodes farther away from the mobile sink consume more energy. Therefore, this paper proposes a distributed network model to collect data based on LEACH architecture.

It can be clearly seen from Figure 1 that all sensor nodes are divided into different clusters. There is a sensor node in each cluster as the cluster head to collect and process the monitoring data of the sensor nodes in the cluster. Then, the cluster heads send the data to the mobile sink. Finally, the mobile sink sends data to the control center and users via satellites and Internet.

Moreover, six assumptions related to model of WSNs are as follows:(1)Position of each sensor node with unique ID is fixed once deployed.(2)Sensor nodes are equipped with GPS device and are location-aware.(3)Each sensor node has limited energy, and their initial energy is the same.(4)The propagation channels are symmetric, i.e., two nodes can communicate using the same transmission power.(5)The mobile sink has unlimited energy, powerful information processing ability, data storage capacity and can move according to a specific motion model.(6)Each sensor node has a fixed number of transmission power levels, and there is no error in signal transmission [16].

### 2.2. Motion Model of Mobile Sink

Assume that in a two-dimensional monitoring area, the motion model of mobile sink is as following:(1)$$\begin{array}{c} x\left(t+1\right)=\mathrm{Fx}\left(t\right)+\Gamma W\left(t\right) \end{array}$$
(2)$$\begin{array}{c} F=\left[\begin{array}{cccc} 1 & \frac{\sin\left(\omega t\right)}{w} & 0 & -\frac{1-\cos\left(\omega t\right)}{w}\\ 0 & \cos\left(\omega t\right) & 0 & -\sin\left(\omega t\right)\\ 0 & \frac{1-\cos\left(\omega t\right)}{w} & 1 & \frac{\sin\left(\omega t\right)}{\omega }\\ 0 & \sin\left(\omega t\right) & 0 & \cos\left(\omega t\right) \end{array}\right],\Gamma =\left[\begin{array}{cc} \frac{t^{2}}{2} & 0\\ t & 0\\ 0 & \frac{t^{2}}{2}\\ 0 & t \end{array}\right] \end{array}$$
where $x\left(t\right)\;=\;{\left[x\left(t\right)\;\dot{x}\left(t\right)\;y\left(t\right)\;\dot{y}\left(t\right)\right]}^{T}$ is the state of mobile sink at time *t*; $x\left(t\right),\dot{x}\left(t\right),y\left(t\right),\dot{y}\left(t\right)$ are the coordinate and speed of the mobile sink in *x* and *y* direction at time *t*; F is the state transition matrix; Γ is noise coefficient matrix; $W\left(t\right)∼N\left(0,Q\left(t\right)\right)$ is the Gaussian noise with covariance $Q\left(t\right)$; *t* is sampling time; ω is angular velocity.

The mobile sink moves according to the above motion model and collects monitoring data of all sensor nodes [1]. When the mobile sink has collected all the data of sensor nodes, it indicates that the one round is completed. As shown in Figure 2, at the beginning of the rth round, the mobile sink is located at $\left(x\left(r\right),y\left(r\right)\right)$ to start collecting monitoring data. When the mobile sink collects data from all the nodes, the rth round completes. Then, the mobile sink starts the r+1th round data collection, as shown in Formula (3).

As can be seen from the Figure 2, the position of mobile sink can be described as:(3)$$\begin{array}{c} \left[\begin{array}{c} x(r+1)\\ y(r+1) \end{array}\right]=\left[\begin{array}{c} x(r)\\ y(r) \end{array}\right]+\left[\begin{array}{c} \Delta x\\ \Delta y \end{array}\right] \end{array}$$
where, $\left(x\left(r\;+\;1\right),y\left(r\;+\;1\right)\right)$ is the coordinate position of the mobile sink at the beginning of the r+1th round; $\left(\Delta x,\Delta y\right)$ is the position change of the mobile sink from the rth round to the r+1th round.

## 3. CEOMS Algorithm

In this section, this paper selects some nodes nearby each sensor node according to energy consumption model to form neighborhood set of this node and then design energy density function related to this set. Moreover, the threshold of cluster head selector is determined by energy density function and motion performance function which is constructed by motion parameters of mobile sink. Finally, the threshold of cluster heads selector may be adaptive adjusted according to mortality of nodes.

### 3.1. Construction of Energy Density Function Based on Neighborhood Nodes

Assume that *N* sensor nodes are randomly deployed in a monitoring area. $K_{opt}$ is the number of nodes to be selected as cluster heads in each round. The desired percentage of cluster heads against the all nodes can be described as:(4)Popt=KoptKoptNN

#### 3.1.1. Construction of Sensor Node Neighborhood Set

As it is known, the energy consumption of data communication is higher than that of sensing data and data processing. This paper uses first order radio model to described energy consumption of node, as shown in the Figure 3 [30].

Assuming that each packet has *k* bit data, the energy consumption of transmitting *k* bit data $E_{TX}$ and the energy consumption of receiving *k* bit data $E_{RX}$ can be expressed as follows:(5)$$E_{TX}\left(k,d\right)=E_{TX-elec}\left(k\right)+E_{TX-amp}\left(k,d\right)=\left\{\begin{array}{cc} kE_{elec}+k\varepsilon_{fs}d^{2} & d⩽d_{0}\\ kE_{elec}+k\varepsilon_{fs}d^{4} & otherwise \end{array}\right.$$
(6)$$E_{RX}\left(k\right)=E_{RX-elec}\left(k\right)=kE_{elec}$$
where, $E_{TX-elec}$, $E_{RX-elec}$ and $E_{TX-amp}$ are the energy consumption of transmitter, the energy consumption receiver and energy consumption amplifier, respectively. $\varepsilon_{fs}$ and $\varepsilon_{mp}$ are coefficient amplify of free space and coefficient amplify of multi-path, respectively. $E_{elec}$ is the energy consumption of 1 bit data processing related to node, *k* is bits of data, *d* is the distance between transmitter and receiver, $d_{0}$ is the critical communication distance.
(7)d0=εfsεfsεmpεmp

It is clearly that energy consumption exponentially increases with distance between the transmitter and the receiver. Thus, the communication range of the node $s_{i}$ may be constrained lower than $d_{0}$ to save energy of WSNs. So, the neighboring node threshold $T_{s_{i}}\left(s_{j}\right)$ of the node $s_{j}$ described in Formula (8) is introduced to determine whether or not node $s_{j}$ should be put in neighborhood set of the node $s_{i}$.
(8)$$T_{s_{i}}\left(s_{j}\right)=\left\{\begin{array}{cc} s_{j}\in \Theta & d_{ij}⩽d_{0}\\ s_{j}\notin \Theta & d_{ij}>d_{0} \end{array}\right.$$
where, Θ is the neighborhood region of node $s_{i}$, $d_{ij}$ is distance between node $s_{i}$ and node $s_{j}$. Nodes located in the neighborhood region Θ of node $s_{i}$ join neighborhood node set N, as shown in Figure 4.

#### 3.1.2. Construction of Energy Density Function

In order to enhance the probability of nodes with high remaining energy to be assigned as cluster heads, the neighborhood nodes remaining energy rate $f_{e}\left(s_{i}\right)$ is introduced as Formula (9).
(9)fesi=Ersi−EavgErsi−EavgEavgEavg
(10)Eavg=∑sj∈NErsj∑sj∈NErsjn′n′
where, $E_{r}\left(s_{i}\right)$ is the remaining energy of the sensor node $s_{i}$, $n^{\prime }$ is the number of neighborhood nodes of the sensor node $s_{i}$, $E_{avg}$ is the average remaining energy of the neighborhood node of the sensor node $s_{i}$.

To describe relation between energy consumption of cluster head and nodes density inside cluster, the neighborhood nodes density function $f_{\rho }\left(s_{i}\right)$ is introduced here.
(11)fρsi=11n′+1n′+1

Combining $f_{e}\left(s_{i}\right)$ and $f_{\rho }\left(s_{i}\right)$, the energy density function $f_{e\rho }\left(s_{i}\right)$ of node $s_{i}$ can be described by Formula (12).
(12)feρsi=fesi·fρsi=Esi−EavgEsi−EavgEavgEavg·11n′+1n′+1

### 3.2. Construction of Motion Performance Function

Although the position of sensor nodes remains unchangeable once they are randomly deployed, the distance between the sensor node and the mobile sink changes with mobile sink moving, leading to the energy consumption changing of the node, as shown in Figure 5.

In Figure 5, when the location of mobile sink is $\left(x\left(r-1\right),y\left(r-1\right)\right)$, some nodes near the mobile sink will consume less energy during data transmission. When the mobile sink moves to $\left(x\left(r\right),y\left(r\right)\right)$, the relative distance between the sensor node and the base station changes, leading to changes in the energy consumption of the nodes.

This paper introduces relative distance $\Delta d_{s_{i}}$ to describe distance changing between node and mobile sink.
(13)$$\Delta d_{s_{i}}=d\left(r\right)-d\left(r-1\right)$$
(14)$$d\left(r\right)=\sqrt{{\left(x\left(r\right)-x_{s_{i}}\right)}^{2}+{\left(y\left(r\right)-y_{s_{i}}\right)}^{2}}$$
(15)$$d\left(r-1\right)\;=\;\sqrt{{\left(x\left(r-1\right)-x_{s_{i}}\right)}^{2}+{\left(y\left(r-1\right)-y_{s_{i}}\right)}^{2}}$$
where, $\left(x_{s_{i}},y_{s_{i}}\right)$ is the axis position coordinates of the sensor node $s_{i}$. $d\left(r\right),d\left(r-1\right)$ are shown in Figure 6.

Moreover, the motion performance function $f_{d}\left(s_{i}\right)$ described in Formula (16) is also introduced to normalize $\Delta d_{s_{i}}$.
(16)fdsi=arctanΔdsiarctanΔdsiππ22ππ22

It is found that $f_{d}\left(s_{i}\right)$ is positive relation with $\Delta d_{s_{i}}$, thus $f_{d}\left(s_{i}\right)$ will increase with $\Delta d_{s_{i}}$, vice versa.

### 3.3. Construction of Adaptive Adjustment Function

Now, $f_{e\rho }\left(s_{i}\right)$, $f_{d}\left(s_{i}\right)$ and $P_{opt}$ are combined to construct initial cluster head selection threshold $T^{\prime }\left(s_{i}\right)$ described in Formula (17).
(17)$$T^{\prime }\left(s_{i}\right)\;=\;\alpha P_{opt}\;+\;\beta f_{e\rho }\left(s_{i}\right)+\gamma f_{d}\left(s_{i}\right)$$
where, α,β,γ are the weight parameters of $T^{\prime }\left(s_{i}\right)$.

#### 3.3.1. Data Transmission

Once initial cluster head selection threshold $T^{\prime }\left(s_{i}\right)$ is determined, the LEACH protocol is used to cluster the nodes and to transmit data. The details are shown in Figure 7.

It can be clearly seen in Figure 7, the entire process of cluster construction and data transmission is divided into four stages. In the first stage, multiple sensor nodes are randomly deployed in the WSNs monitoring area, and the positions of the nodes remain unchanged, as shown in Figure 7a.

In the second stage, $T^{\prime }\left(s_{i}\right)$ of each node should be compared with $T_{rand}\left(s_{i}\right)$, which is a rand number uniformly distributed random in the [0, 1]. Then, described in Formula (18), if $T_{rand}\left(s_{i}\right)<T^{\prime }\left(s_{i}\right)$, then cluster head indicator $T_{c}^{\prime }$ related to this node was set to be 1 and put this node into cluster head set C, else cluster head indicator $T_{c}^{\prime }$ related to this node was set to be 0 and put this node into non-cluster head set $C^{\prime }$. The details are shown in Figure 7b.
(18)$$T_{c}^{\prime }=\left\{\begin{array}{cc} 1 & T_{rand}\left(s_{i}\right)<T^{\prime }\left(s_{i}\right)\;\\ 0 & T_{rand}\left(s_{i}\right)⩾T^{\prime }\left(s_{i}\right) \end{array}\;\right.$$

In the third stage, each cluster head broadcasts a message to all nodes, and each node chooses to join corresponding clusters according to the strength of received signal, as shown in Figure 7c. When all sensor nodes join the cluster, each sensor node informs its selection through the carrier sense multiple access (CSMA) MAC protocol. Moreover, mobile sink can act as cluster heads of some nodes nearby mobile sink, and then each cluster head creates a time division multiple access (TDMA) schedules for its cluster members.

In the fourth stage, the data transmission process of WSNs is shown in Figure 7d. The sensor nodes generate monitoring data in each round and send data to the cluster heads within the allocated transmission time. Cluster members are dormant at the non-allocated transfer time. While the cluster heads are always active to collect data of all nodes in the cluster. Once the cluster head collects all the data completely, it would fuse these data and then forward the fused data to the mobile sink. The current round will be over once all data are collected by the mobile sink completely.

#### 3.3.2. Construction of Adaptive Adjustment Function Based on Node Death Percentage

Since energy consumption of the sensor nodes increase with working time of WSNs, leading to surviving sensor nodes reducing. Thus, parameters related to cluster head selection threshold should be adjusted to balance energy of WSNs. This paper embeds an adjustment function $g\left(P_{dead}\right)$ described in Formula (20) into the sigmoid function [31] described in Formula (19).
(19)y=111+e−x1+e−x
(20)gPdead=111+e20Pdead−101+e20Pdead−10
where,
(21)Pdead=NdeadNdeadNN
here, $P_{dead}$ is the percentage of nodes death; $N_{dead}$ is the number of nodes death; *N* is the total number of sensor nodes. It is found that the number of node deaths increases with $P_{dead}$. The curve of $g\left(P_{dead}\right)$ is shown in Figure 8.

Adaptive cluster head selection threshold $T(s_{i})$ described in Formula (22) can be designed by combining $P_{opt}$, $f_{e\rho }\left(s_{i}\right)$, $f_{d}\left(s_{i}\right)$ and $g\left(P_{dead}\right)$.
(22)$$T\left(s_{i}\right)=g\left(P_{dead}\right)\left[\alpha P_{opt}\;+\;\beta f_{e\rho }\left(s_{i}\right)+\gamma f_{d}\left(s_{i}\right)\right]$$

Observing Formula (22), it can be found that the adaptive cluster head selection threshold $T(s_{i})$ is positively correlated with the desired percentage of cluster head $P_{opt}$, the motion performance function $f_{d}\left(s_{i}\right)$ and the energy density function $f_{e\rho }\left(s_{i}\right)$. As the network runs, the cluster head selection threshold $T(s_{i})$ decreases with the increase of the number of dead nodes. Moreover, when the construction of the adaptive cluster head selection threshold is completed, the adaptive cluster head selection threshold $T(s_{i})$ is calculated for all sensor nodes in the network to select the cluster head in each round. Then, nodes selected as the cluster head are added to the cluster head set C, and the nodes not selected as the cluster head are added to the non-cluster head set $C^{\prime }$.

The symbolic instructions in the adaptive cluster head selection threshold building steps above are shown in Table 1.

### 3.4. CEOMS Algorithm

From above description, it can be concluded that proposed CEOMS algorithm combines underlying factors related to energy balance WSNs including energy, density of nodes and motion parameters of mobile sink to adaptive adjust threshold of cluster head selection. The flow chart of CEOMS algorithm is shown in Figure 9.

Pseudo code of the CEOMS algorithm is shown in the Algorithm 1.
**Algorithm 1** CEOMS algorithm**1:** Initialization, set $r_{\max}$ ( Maximum number of rounds of the algorithm)**2:** **for**$r=1:r_{\max}$**do****3:**   Calculate neighborhood nodes threshold $T_{s_{i}}\left(s_{j}\right)$ according to Formula (7)**4:**   **if**$s_{j}\in \Theta$**then****5:**      $s_{j}$ joins the neighborhood set *N* of node $s_{i}$**6:**   **end if****7:**   **for**$\forall s_{j}\in N$**do****8:**     Calculate energy density function $f_{e\rho }\left(s_{i}\right)$ according to Formula (11)**9:**   **end for****10:**   Calculate motion performance function $f_{d}\left(s_{i}\right)$ according to Formula (14)**11:**   Calculate initial cluster head selection threshold $T^{\prime }\left(s_{i}\right)$ according to Formula (15) and (3)**12:**   Calculate adaptive adjustment function $g\left(P_{dead}\right)$ according to Formula (18)**13:**   Construct the adaptive cluster head selection threshold $T(s_{i})$ according to Formula (20)**14:**   Perform cluster head selection and data transfer**15:**   **if**$r<r_{\max}$**then****16:**    r=r+1, return to Step 3**17:**   **end if****18:** **end for**

## 4. Simulation Results and Analysis

### 4.1. Experimental Parameters

Assume that WSNs composed of *N* sensor nodes were randomly distributed in the monitoring area $S_{M}$. The initial position of the mobile sink was located in the monitoring area with coordinates (0 m, 0 m). The moving trajectory of mobile sink is shown in Figure 10.

Simulation parameters related to the experiment are shown in Table 2.

### 4.2. Simulation Results and Analysis

This paper used some indicators including survival time of network nodes, total remaining energy, and the balance of network energy consumption to verify the feasibility and effectiveness of the proposed CEOMS algorithm by comparative experiment related to some algorithms such as the ILEACH algorithm [23], DBCH algorithm [24] and LEACH-DT algorithm [25]. For a fair comparison, the performances of the four algorithms above were compared under the same environment conditions using MATLAB2019a equipped with Windows 10-64bit on Intel(R) Core (TM) i5-6500H CPU and 8 GB RAM.

#### 4.2.1. Survival Time Analysis of Network Nodes

When the monitoring data of all sensor nodes were collected, it indicated that one round had been completed, and then the next round would be entered. In each round, different algorithms had different energy consumption optimization strategies. Therefore, in each round, the remaining energy of the nodes was different, resulting in a different number of surviving nodes. The variation trends of survival nodes of the four algorithms are shown in Figure 11.

It can be seen from Figure 11 that there were 100 survival nodes at the beginning of the network. With the continuous running of the network, the corresponding curves of the four algorithms all showed a downward trend. With the increase of energy consumption, the number of surviving nodes decreased and death nodes gradually appear. The round of first death node with respect to ILEACH, DBCH, LEACH-DT and CEOMS algorithm were 99th, 73th, 104th and 144th, respectively. Compared with the other three algorithms, first death node time of CEOMS algorithm was extended by 45.4%, 97.3% and 38.5%, respectively. The round of all death nodes with respect to ILEACH, DBCH, LEACH-DT and CEOMS algorithm were 1111th, 1225th, 1199th and 1645th, respectively. Compared with the other three algorithms, the life cycle of the CEOMS algorithm extended by 48.1%, 34.2% and 37.1%, respectively. The relationship between the number of node deaths and the number of running rounds of the four algorithms is shown in Figure 12.

As can be seen from the above Figure 12, compared with the other three algorithms, the CEOMS algorithm took into account the remaining energy of nodes and the mortality of nodes when selecting cluster heads, so CEOMS algorithm had the longest working time and could effectively extend the network life cycle.

#### 4.2.2. Analysis of Total Remaining Energy of Nodes

With the running of WSNs, the total remaining energy of WSNs reduced gradually. Total remaining energy of all nodes of the four algorithms versus the rounds increasing are plotted in Figure 13.

In this figure, the number of nodes was set to be 100 and initial energy of each node was set to be 1 J. It can be found that the curve corresponding to CEOMS algorithm was higher than other three algorithms, which indicated that the total residual energy of WSNs corresponding to CEOMS algorithm was greater than that of other three algorithms. When the total remaining energy of the WSNs was 0, the round of running of ILEACH algorithm, DBCH algorithm, LEACH-DT algorithm and CEOMS algorithm was 1111th, 1225th, 1199th and 1645th, respectively, which indicated the entire network running time of CEOMS algorithm was the longest. Different from the other three algorithms, the CEOMS algorithm took the average remaining amount of neighborhood nodes and the current remaining energy as key factors when selecting cluster heads. So, the CEOMS algorithm could effectively save the network energy and prolong network lifetime.

#### 4.2.3. Comparative Analysis of Remaining Energy Distribution of Network Nodes

It is clear that remaining energy of the node was in relation to the position of the mobile sink. In the 300th round, the difference of remaining energy of each node between four algorithms and classic LEACH is shown in Figure 14, respectively.

It can be found that WSNs which used CEOMS algorithm had a larger surface fluctuation, which meant using the CEOMS algorithm could save the energy of the nodes. Moreover, at the 100th, 300th, 500th, 800th and 1000th round, the percentage of energy saving related to the four algorithms is described in the Figure 15.

It can be found that the curve corresponding to the CEOMS algorithm was higher than other three curves. It also demonstrates that the CEOMS algorithm had superior in energy saving.

#### 4.2.4. Analysis of Variance of Nodes Remaining Energy

In WSNs, static sink node will lead to excessive load and unbalanced energy consumption of nodes near the sink node. The proposed CEOMS algorithm introduced mobile sink and considered the influence of motion parameters on the selection of cluster heads to design motion performance function to alleviate the imbalance of energy consumption. Now, variance of nodes remaining energy (Figure 16) was introduced here to describe the balance degree of WSNs.

Figure 16 shows that the initial remaining energy variance of four algorithms was 0, which meant the initial energy distribution of WSNs was entirely uniform. With the running of WSNs and changing of mobile sink location, the curve corresponding to CEOMS algorithm was always under the curves corresponding to other three algorithms. Moreover, variance of residual energy of ILEACH, DBCH, LEACH-DT and CEOMS algorithm reduced to 0 at the 1111th, 1225th, 1199th and 1645th round, respectively, which indicated the CEOMS algorithm not only effectively balanced network load but also extended network life.

#### 4.2.5. Applicability Analysis of Network Lifetime

In order to further verify the applicability of the proposed CEOMS algorithm to prolong the life cycle of the network, the location of the network nodes was randomly generated for many experiments. Figure 17 shows the network life cycle of the ILEACH algorithm, DBCH algorithm, LEACH-DT algorithm, and CEOMS algorithm when the node position was randomly generated.

Several experiments were performed by randomly generating node positions. From the results of the five experiments shown in Figure 16, the total death times (network life cycles) of nodes in the network corresponding to the CEOMS algorithm were: 1642 rounds, 1671 rounds, 1743 rounds, 1674 rounds and 1697 rounds. Compared with the network life cycle data of the ILEACH algorithm, the DBCH algorithm and the LEACH-DT algorithm, the CEOMS algorithm had the longest network life cycle, indicating that the CEOMS algorithm could effectively extend the network life cycle and had applicability.

## 5. Conclusions

Some factors including remaining energy and density within the neighborhood radius of sensor nodes, the location and velocity of mobile sink and the number of dead nodes may impact on energy balance of WSNs. This paper proposed a novel cluster-based energy optimization algorithm—CEOMS, to select cluster head by comprehensively considering above factors impact on energy balance of WSNs. The proposed algorithm firstly introduced the energy density function by considering the residual energy rate and density within the neighborhood radius of nodes to reduce the randomness of the cluster head selection. Secondly, the motion performance function was designed based on the variables of the motion parameters of the mobile sink, which effectively balanced the network load and reduced the data delay. Finally, an adaptive adjustment function related to node mortality was proposed to adjust the factors of the cluster head selection threshold, which prolonged the network life. The energy density function, motion performance function and adaptive adjustment function worked together to improve the self-adaptability of cluster head, balance network load, reduce the data delay and prolong the network life cycle.

In addition, the proposed algorithm only uses a mobile sink, which will lead to partial data loss and delay when the monitoring area is large. Therefore, in the future work, how to apply multiple mobile sink to collect data in WSNs should be taken into consideration [32]. 

## Figures and Tables

**Figure 1 sensors-21-02523-f001:**
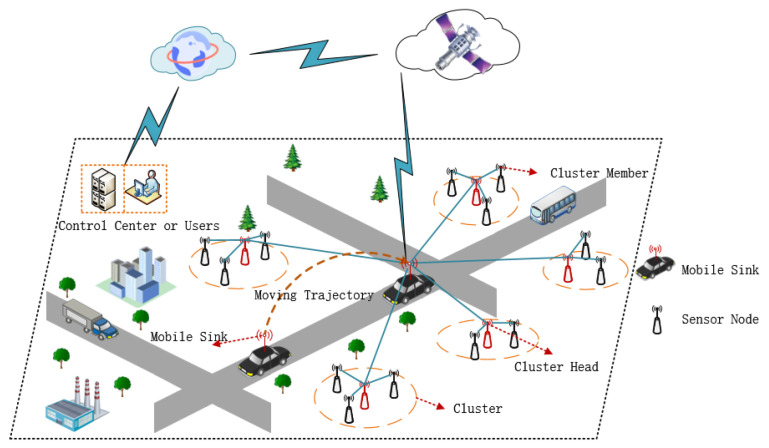
WSNs architecture.

**Figure 2 sensors-21-02523-f002:**
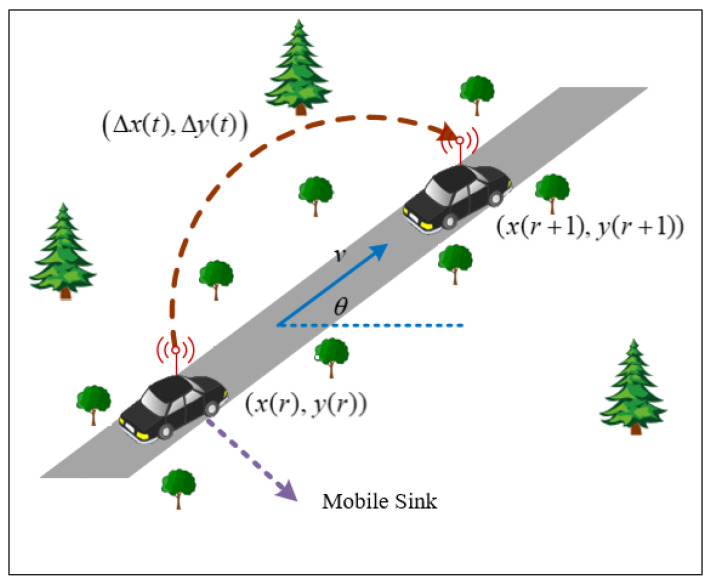
Motion model.

**Figure 3 sensors-21-02523-f003:**
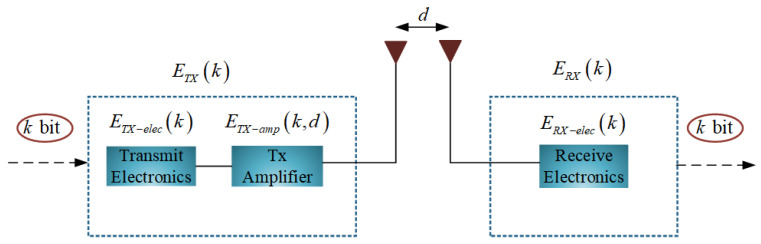
First order radio model.

**Figure 4 sensors-21-02523-f004:**
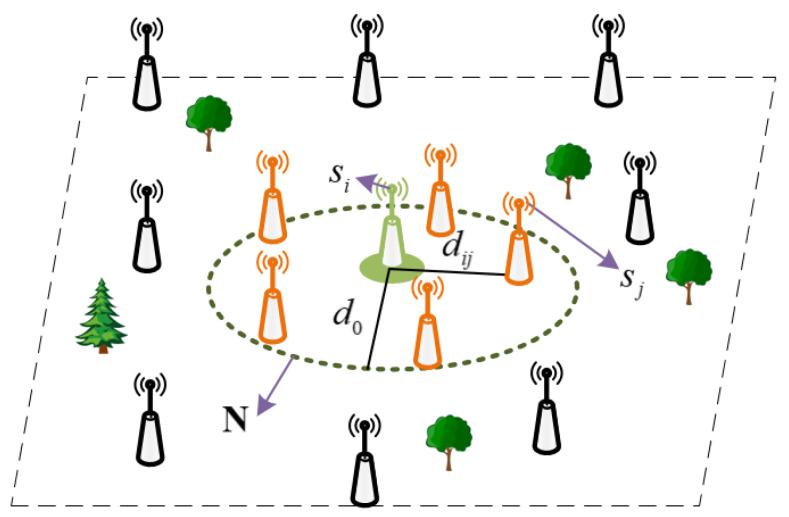
Neighborhood node set.

**Figure 5 sensors-21-02523-f005:**
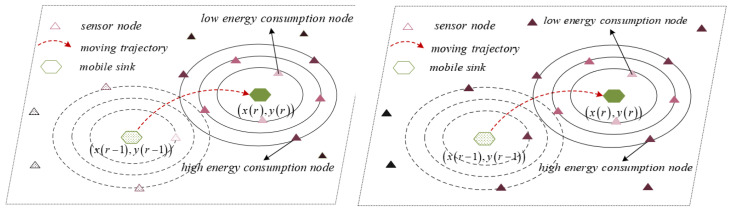
Energy intensity distribution.

**Figure 6 sensors-21-02523-f006:**
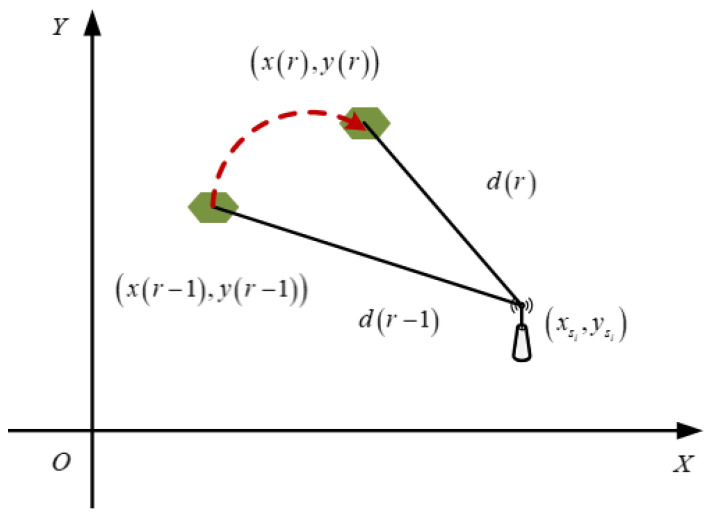
Change of relative distance.

**Figure 7 sensors-21-02523-f007:**
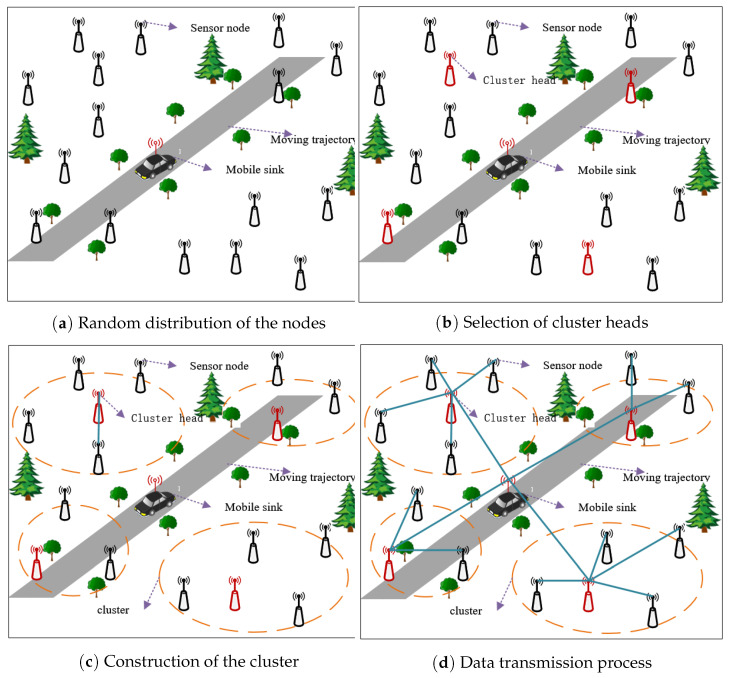
Cluster construction and data transmission.

**Figure 8 sensors-21-02523-f008:**
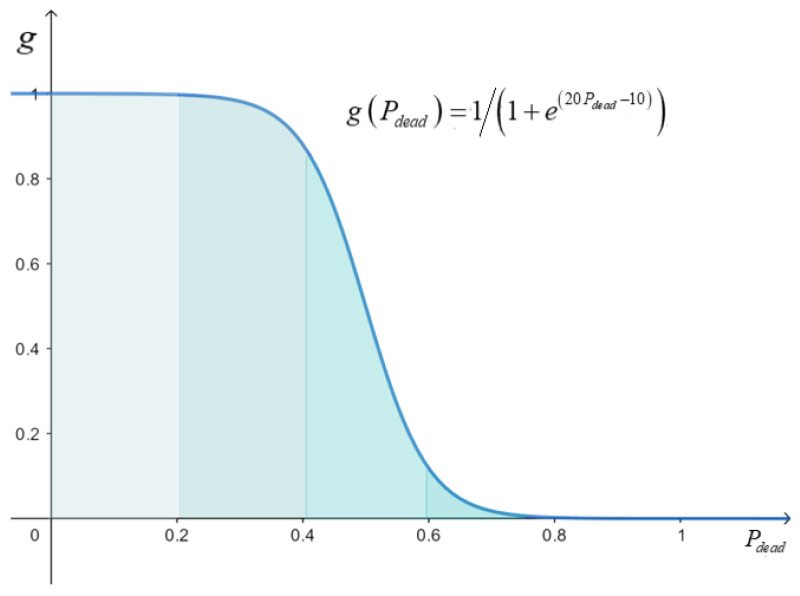
Curve of adaptive adjustment function.

**Figure 9 sensors-21-02523-f009:**
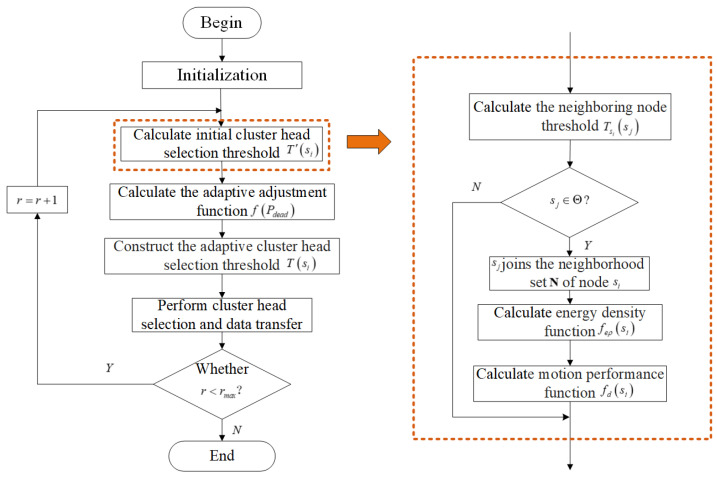
Cluster-Based Energy Optimization with Mobile Sink (CEOMS) algorithm flow chart.

**Figure 10 sensors-21-02523-f010:**
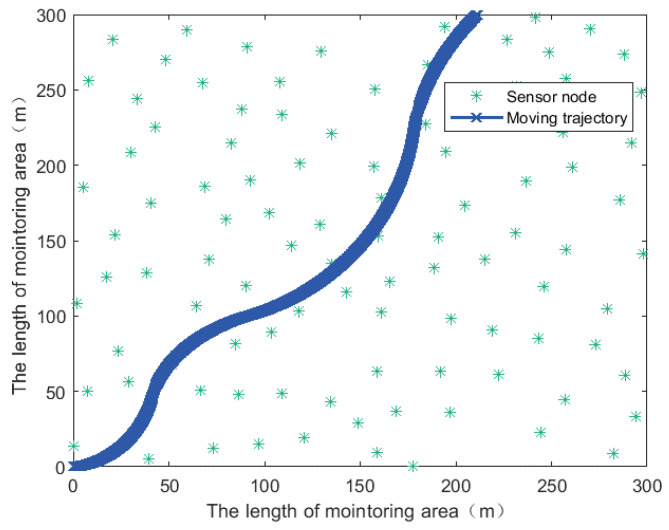
Moving trajectory of the mobile sink.

**Figure 11 sensors-21-02523-f011:**
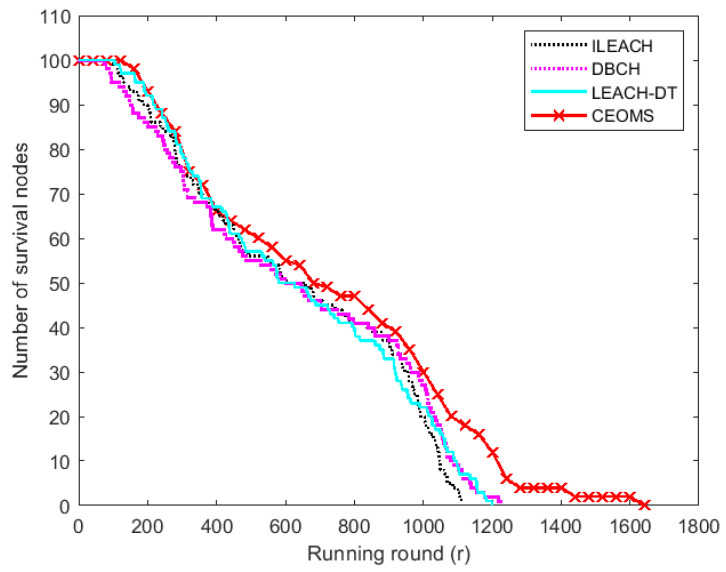
The number of surviving nodes.

**Figure 12 sensors-21-02523-f012:**
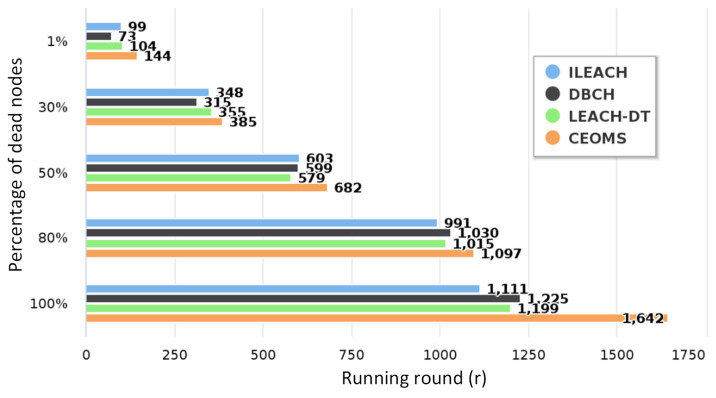
The relationship between node death ratio and the number of rounds.

**Figure 13 sensors-21-02523-f013:**
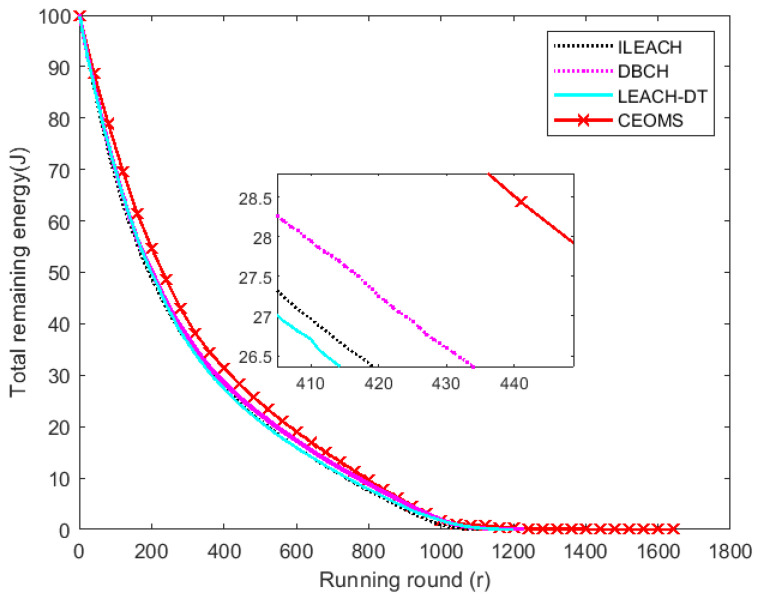
Change trend of total residual energy.

**Figure 14 sensors-21-02523-f014:**
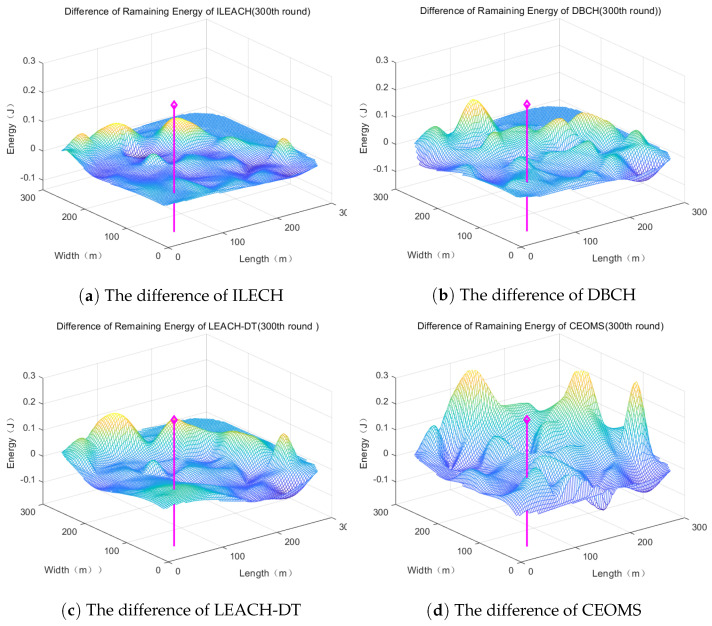
The difference of remaining energy distribution.

**Figure 15 sensors-21-02523-f015:**
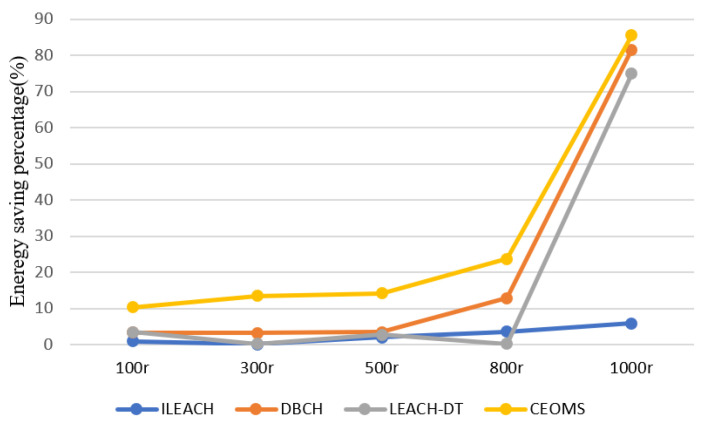
Energy saving percentage of four algorithms.

**Figure 16 sensors-21-02523-f016:**
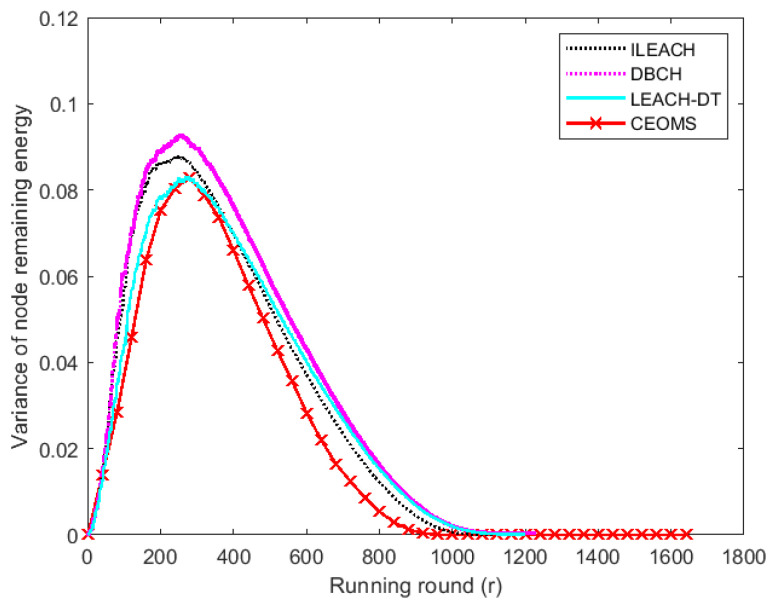
The variance of remaining energy of four algorithms.

**Figure 17 sensors-21-02523-f017:**
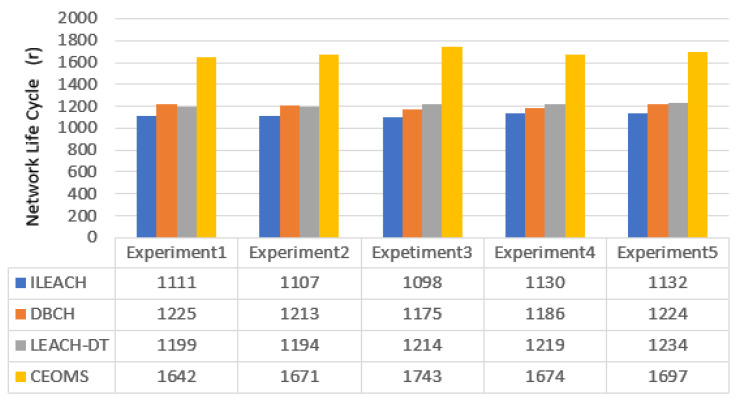
Network life cycle of the four algorithms.

**Table 1 sensors-21-02523-t001:** Symbolic representation in adaptive cluster head selection threshold construction steps.

Symbol	Definition	Symbol	Definition
$P_{opt}$	Desired Percentage of Cluster Head	$T_{s_{i}}\left(s_{j}\right)$	Neighboring Nodes Threshold
$f_{e}\left(s_{i}\right)$	Neighborhood Nodes Remaining Energy Rate	$f_{e\rho }\left(s_{i}\right)$	Neighborhood Nodes Density Function
$f_{\rho }\left(s_{i}\right)$	Neighborhood Nodes Density	$f_{d}\left(s_{i}\right)$	Motion Performance Function
$T^{\prime }\left(s_{i}\right)$	Initial Cluster Head Selection Threshold	α,β,γ	Weight Parameters of $T^{\prime }\left(s_{i}\right)$
$T_{rand}\left(s_{i}\right)$	Rand Number Uniformly distributed in [0, 1]	$P_{dead}$	Percentage of Nodes Death
$N_{dead}$	Number of Nodes Death	*N*	Total Number of Nodes
$g\left(P_{dead}\right)$	Adjustment Function	$T(s_{i})$	Adaptive Cluster Head Selection Threshold

**Table 2 sensors-21-02523-t002:** Experimental parameters.

Name	Symbol	Value
Area of Monitor	$S_{M}$	300 m × 300 m
Number of Sensor Nodes	*N*	100
Initial Position of Mobile sink	$\left(x_{MS},y_{MS}\right)$	(0 m, 0 m)
Angular Velocity	ω	0.0025, −0.0025, 0.0019, −0.0019 rad/s
Initial Energy of Node	$E_{0}$	1J
Communication Energy Consumption	$E_{TX}/E_{RX}$	50 nJ/bit
Energy Consumption of Signal Amplification in Free Space	$\varepsilon_{fs}$	10 pJ/bit/m${}^{2}$
Energy Consumption of Signal Amplification in Multipath	$\varepsilon_{mp}$	0.0013 pJ/bit/m${}^{2}$
Energy Consumption of Data Fusion	$E_{\mathrm{DA}}$	5 nJ/bit/packet
Sampling Rate of Sensor Nodes	$R_{d}$	Accumulative every 3 s for 5 min
Length of Control Signal	$l_{1}$	100 bit
Length of Monitor Data	$l_{2}$	4000 bit
Weight Parameters of $T^{\prime }\left(s_{i}\right)$	α,β,γ	0.85, 1, 0.2
Optimal Probability of Cluster Heads	$P_{opt}$	0.1
Maximum number of running rounds	$r_{max}$	1800

## Data Availability

The data presented in this study are available on request from the corresponding author.

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
