# Peer review of "A Cluster-Based Energy Optimization Algorithm in Wireless Sensor Networks with Mobile Sink"

_sensors, 2021, doi:10.3390/s21072523_

Round 1
Reviewer 1 Report
The authors propose a cluster-based energy optimization algorithm called CEOMS (Cluster-based Energy Optimization with Mobile Sink).
The paper is well written and it is scientifically sound. The authors need to address the following minor issues before the publication:
1- The Introduction has to be split into Introduction and related work.
2- Figure 1 for instance bring the idea of a vehicular network. What would be a cluster member? A parked vehicle? A road side infrastructure communication device? Please, explain.
Author Response
Dear Ms. Jane Xu and reviewers,
Thanks very much for taking your time and effort to review this manuscript. Your careful review has helped to make our study clearer and more comprehensive. Here we express our heartfelt thanks! We have studied comments carefully and have made correction which we hope meet with approval.
We would like to thank you for allowing us to resubmit a revised copy of the manuscript and we highly appreciate your time and consideration. We wish good health to you, your family, and community.
Kind regards,
Ms. Bai

Reviewer 2 Report
Firstly, the paper requires extensive editing of English language and style.
Remaining comments:
- Lines 105-108. Three assumptions on the WSN has been taken. What about following assumptions:
- Maximum power of signals generated by the sensors (it influences on communication range of the sensor motes); or at least communication range; note that it can change during sensor’s operation because of the energy consumption;
- Is the transmission power adjusted? If yes, the communication range will change.
- The same assumption is valid for a mobile sink.
- Is any knowledge of the sensors positions in the system?
- What about the signal propagation model assumption?
- Do you assume errors in signaling transmission?
- Lines 109-110. “…position of mobile sink is fixed when it collects data of all nodes.” Does it mean that the sink is able to communicate with all nodes (Cluster Heads) in the WSN during the fixed position? This is unrealistic assumption because of the limited radio communication range.
- The sink motion model is illegible in my opinion, especially eq. 1 (and its description); it operates firstly on time, then on round; what do you mean by state transition matrix? What does the round mean; is it connected with time?
- Lines 117. “The mobile sink broadcasts its location to WSNs and collects monitoring data of all senor nodes.” – Are there more WSNs in the area of operation? Assumption was that sensors are randomly distributed in the area of operation and they belong to one WSN. Collection of monitoring data from all sensor motes can be difficult in a general sense.
- Lines 140-142. The description is unclear; must be corrected according to the source of the energy consumption model; the energy consumption model depends on the propagation model that also should be assumed and clarified.
- Line 149-150. What is difference between N and Q? Later, N does mean all nodes.
- Figure 4. What is a difference between the two pictures presented in this Figure?
- Line 179. Weight parameters should be clarified here.
- Section 3.4. The calculations should be done for each node (i.e. si). It should be written somewhere in the text.
- Section 4. I suggest using “Running round” instead of “Run time” in the labels of the “x” axis in the Figures.
- To increase results credibility, the simulations (calculations) should be repeated many times with changed initial random numbers settings. It is difficult to say that the differences between the plots are relevant without statistical analysis.
- I suggest commenting the figures not only saying that the CEOMS algorithm is better, but to explaining what the reason of such behavior is.
- In conclusion (section 5). Please consider explaining: future work, difficulties in a practical application of the algorithm, influence of the simplifications on the results.
Author Response
Please see the attachment “Response to Reviewer 2”.

Reviewer 3 Report
Please find below the summary of comments:
- In line 7 of the abstract, “senor nodes being” should be corrected to “sensor nodes being”.
- In line 7 of the abstract, “based on LEACH architecture” should be replaced with “based on Low Energy Adaptive Clustering Hierarchy Protocol (LEACH) architecture”.
- In the abstract authors should briefly explain what is meaning of “network survival nodes”. Does it refer to non-Cluster head nodes? Or have not yet joined any clusters?
- In the Introduction, the relation of these functions (energy density function, motion performance function and adaptive adjustment function) should be clearly stated. Are they independent? What is their effect on each other?
- It would be better for the authors to study their related works so that their contributions can be better seen by analyzing their strengths and weaknesses. Why is the separation section not mentioned as related work?
- Which real-world application can be mapped to the arcchitecture presented in Figure 1? What kind of data do we collect in this architecture? What is the volume of data?
- All calculations (CH selection, etc.) are done in which part of the architecture?
- Is the method centralized or distributed? Do not need a powerful static sink?
- What are the medium access control (MAC) between the mobile sink and the nodes?
- Do all nodes have access to each other? How can Authors ensure that the number of cluster heads does not reduce efficiency with the number of isolated nodes?
- The mobile sink as a car in the manuscript should stop (park) while collecting data. What is the effect of the park time on end-to-end delay network? Has this effect been seen?
- In 2.2. Motion Model of Mobile Sink: Why do we need to design a new motion model for mobile sink? Why are the current methods not enough?
- In Section 3.2. Construction of Motion Performance Function: Is the velocity of the mobile sink fixed?
- What is the meaning of “Node Mortality”?
- As I understand it “Pdead” is the probability of node death. What is your probability of stopping the algorithm completely?
- A big number of variables are used that it is better to use the notation table.
- In Table 1, it is better to express the values of communication range and sensing rate of nodes.
- What is the meaning of “Survival Time Analysis of Network Nodes” and how is it calculated?
Author Response
Please see the attachment "Response to Reviewer 3".

Round 2
Reviewer 2 Report
Dear Authors, I accept all the clarifications. Thank you. My concerns have been addressed. Maybe conclusions are a bit general. I recommend the paper for publication.
Author Response
Please see the “Response to Reviewer 2(Round 2)”

Reviewer 3 Report
Authors should check the paper in terms of grammar and typos. The paper technically sounds, but still, there are few typos that should be fixed.
Author Response
Please see the “Response to Reviewer 3(Round 2)”
